# Projecting the Global Potential Geographical Distribution of *Ceratitis capitata* (Diptera: Tephritidae) under Current and Future Climates

**DOI:** 10.3390/biology13030177

**Published:** 2024-03-08

**Authors:** Jiawei Rao, Yu Zhang, Haoxiang Zhao, Jianyang Guo, Fanghao Wan, Xiaoqing Xian, Nianwan Yang, Wanxue Liu

**Affiliations:** 1State Key Laboratory for Biology of Plant Diseases and Insect Pests, Institute of Plant Protection, Chinese Academy of Agricultural Sciences, Beijing 100193, China; 18702676450@163.com (J.R.); zhangyuu960606@163.com (Y.Z.); hx_zhao@bjfu.edu.cn (H.Z.); guojianyang@caas.cn (J.G.); wanfanghao@caas.cn (F.W.); liuwanxue@caas.cn (W.L.); 2Institute of Western Agriculture, Chinese Academy of Agricultural Sciences, Changji 831100, China

**Keywords:** *Ceratitis capitata*, CLIMEX, potential geographical distribution, climate change

## Abstract

**Simple Summary:**

*Ceratitis capitata* is a globally destructive pest distributed mainly in the Mediterranean, African, and South American regions that causes significant economic losses. In this study, we adjusted the biological parameters of *C. capitata* and used the CLIMEX model to predict its potential geographical distribution under current and future conditions. Under the current climatic conditions, southern Asia, southern Oceania, southern North America, southern and central South America, and mainly southern and central Africa are highly favorable habitats. Owing to changes in temperature under future climatic conditions, the area of geographical distribution is projected to decrease and shift to higher latitudes.

**Abstract:**

The Mediterranean fruit fly, *Ceratitis capitata* (Wiedemann), which is native to tropical Africa, has invaded more than 100 countries and constitutes a risk to the citrus sector. Studying its potential geographical distribution (PGD) in the context of global climate change is important for prevention and control efforts worldwide. Therefore, we used the CLIMEX model to project and assess the risk of global invasion by *C. capitata* under current (1981–2010) and future (2040–2059) climates. In the prevailing climatic conditions, the area of PGD for *C. capitata* was approximately 664.8 × 10^5^ km^2^ and was concentrated in South America, southern Africa, southern North America, eastern Asia, and southern Europe. Under future climate conditions, the area of PGD for *C. capitata* is projected to decrease to approximately 544.1 × 10^5^ km^2^ and shift to higher latitudes. Cold stress was shown to affect distribution at high latitudes, and heat stress was the main factor affecting distribution under current and future climates. According to the predicted results, countries with highly suitable habitats for *C. capitata* that have not yet been invaded, such as China, Myanmar, and Vietnam, must strengthen quarantine measures to prevent the introduction of this pest.

## 1. Introduction

Fruit flies of the family Tephritidae comprise approximately 4300 recognized species distributed across approximately 500 genera globally [1]. They are widely distributed agricultural pests that infest diverse fruits and vegetables, causing serious economic losses [2,3,4,5]. The Mediterranean fruit fly (MFF), *Ceratitis capitata* (Wiedemann) (Diptera: Tephritidae), is the most widely distributed and serious pest species in this family. The MFF is native to tropical Africa [6] and was discovered in Spain at the beginning of the 19th century [7], followed by various other regions of the world [8]. It is currently distributed throughout the Mediterranean, much of Africa, and the Middle East, including the Indian Ocean islands, South and Central America, western Australia, and the Pacific between 1989 and 2014 [9]. The spread of the MFF is primarily attributed to increased human mobility and international commerce [10]. The European and Mediterranean Plant Protection Organization listed it as an A2 pest, and the United States, New Zealand, Belarus, Moldova, and Mexico listed this species as a quarantine pest [11]. This species exhibits a high degree of polyphagy, with a diet comprising more than 300 host plants. It demonstrates adaptability to diverse climates, as evidenced by its ability to thrive in various environments [12]. The MFF can also transmit fruit-rotting fungi, causing fruit to rot and fall off trees [13]. The MFF larvae leave a large hole in the fruit, which serves as a potential gateway for fungi and bacteria. Moreover, by the time the larvae leave the fruit, the fruit has suffered from infestation. The gateway is made by the female when she lays the eggs [14]. In Brazil, this pest is estimated to have caused 20–50% of total production losses, while its invasion in California has resulted in an estimated USD 3.6 billion in production losses [9]. The enormous economic losses caused by this pest have raised concerns worldwide.

Global warming-induced climate change accelerates insect growth rates, thereby accelerating changes in insect population dynamics as well as altering patterns of population distribution and shifting the range of insects to higher latitudes, including polar regions [15,16,17,18]. A report from the Intergovernmental Panel on Climate Change indicated that under multiple greenhouse gas emission scenarios, the global average temperature is projected to increase by 2–4 °C by the end of the 21st century. Notably, MFF is a holometabolous insect with four distinct life stages, and its growth and development are closely related to environmental temperatures [17]. For example, an increase in temperature can expand their overwintering area and increase their survival rate [16]. Additionally, under elevated temperatures, larvae may undergo pupation at an accelerated rate, but lower survival of the MFF in the immature stage has been observed at 35 °C [19]. Therefore, it is essential to ascertain the potential geographical distribution (PGD) of the MFF under the conditions of anticipated climate change to effectively monitor and control this pest on a global scale. 

Species distribution models (SDMs) are predictive instruments illustrating the impact of environmental changes on species distribution [14]. The CLIMEX model has been widely used to predict the abundance and PGD of organisms according to biological parameters and climate data [20,21,22]. The main advantages of the CLIMEX model are that it can be theoretically calibrated based on geographical distribution data and biological properties [17]. Moreover, biological data can be continually revised and incorporated into the CLIMEX model, thereby enhancing accuracy. Many studies have used CLIMEX to predict the PGD of tephritid flies. However, these studies focused on the potential distribution of the MFF in the current climate and did not predict its global distribution under future climatic conditions [18,19,20,21].

Thus, in this study, we used the CLIMEX 4.0.0 model based on relevant biological parameters and climate data with the aim of analyzing the following: (1) the PGD of MFF under current and future climates; (2) the impact of climate change on the PGD; (3) the spatial variation in PGD; and (4) the impact of individual climate factors on the PGD under current and future climatic conditions. This information can serve as a valuable reference for decision makers regarding the integrated management of the MFF.

## 2. Materials and Methods

### 2.1. CLIMEX Model

CLIMEX is a bioclimatic model that predicts the PGD of a species based on its biological parameters and climate data [18]. In this study, we used the ‘Compare Location’ function of CLIMEX 4.0.0 (Hearne Science Software, South Yarra, Australia) for predicting PGD. The model evaluates the climatic appropriateness of a species for a given location by calculating the ecoclimatic index (EI). Theoretically, the EI ranges from 0 (indicating unsuitability) to 100 (indicating year-round suitability). Scores of 100 are typically attained only in regions characterized by high climatic stability, such as certain equatorial zones. The EI value was determined based on calculations involving the annual growth index (GIA), stress index (SI), and stress interaction index. For more detailed information on the computational formulas, refer to Kriticos et al. [19].

### 2.2. Species Occurrence Records

The global distribution of the MFF was recorded from the Centre for Agriculture and Biosciences International database (131 records), Global Biodiversity Information Facility (6295 records), and Barcode of Life Data System (51 records) [11,20,21]. The biological credibility of the data was validated, and duplicates as well as records lacking geographic coordinates were eliminated from the database. Finally, 1845 records were retained and used for parameter fitting. These records are geographically representative of the current global distribution of the MFF (Figure 1).

### 2.3. Climate Data

The current climate data were obtained from the CM30_1995H dataset in CliMond (these climate data are provided by the European Centre for Medium-Range Weather Forecasts and are from 67,420 distributed meteorological stations worldwide) [22], which include environmental variables such as monthly average rainfall and temperature, daily temperature difference, and vapor pressure from 1981 to 2010. Based on these variables, the average daily maximum and minimum temperatures and the average monthly relative humidity from 09:00 to 15:00 were calculated for 1981–2010. Access 1.0, a global climate model developed by the Australian Bureau of Meteorology, was used to simulate the potential distribution of the MFF under future climate conditions for the period from 2040 to 2059. In the future climate scenario, by the mid-century, temperatures are projected to increase by 2 °C. The warmer atmosphere can hold more water vapor, leading to a corresponding increase in humidity. Consequently, in this scenario, there is an anticipated increase in extreme rainfall events, such as those classified as “once-in-ten-years” precipitation events, with the likelihood of such events rising to 71.5% [23]. The choice of climate period is based on the climate data available on the CliMond climate website (dedicated to CLIMEX current and future climate data) and defaults to current climate (1981–2010) and future climate (2040–2059).

### 2.4. Irrigation Data

The MFF is a tropical species capable of surviving in temperate climates. In particular, there have been MFF outbreaks in California in the United States. Under natural rainfall conditions, California summers are too dry to support sufficient host development, leading to a decline in MFF populations. Thus, the MFF is likely to persist only in areas with irrigated crops or adequate soil moisture [24]. In order to maximize the fit of our predictions to the actual distribution of the MFF, the scenario of 1.5 mm day^−1^ of summer irrigation [25] from CLIMEX was applied.

### 2.5. Fitting CLIMEX Parameters

Previous studies have predicted the PGD of the MFF by using the CLIMEX model [24,26], which lists MFF parameters as an example [27]. However, these predictions did not encompass all documented distributions of MFF, most likely because records of the known occurrence of this species have been recently updated. Therefore, the parameter values required adjustment based on the new occurrence data. In this study, we first used the parameters of Vera et al. [24] and Szyniszewska et al. [9] as the initial values of the iterative tuning parameter and carried out a preliminary fit. We found that some high latitudes in parts of Africa are non-viable areas, but regardless of this, they have an established population distribution. Thereafter, the temperature index and degree days per generation parameter values were determined from the literature related to the MFF, followed by a determination of most of the parameter values of cold, heat, dry, and wet stress according to the CLIMEX Mediterranean template. Considering the host plants of MFF, the irrigation index was added for further tuning of the parameter. Table 1 lists the adjusted final parameter values. 

#### 2.5.1. Growth Index

The CLIMEX model uses the GIA to assess the suitability of a location for population growth, which ranges from 0 to 100. This index typically combines the responses to temperature, soil moisture, associated day length, and species diapause [27].

#### 2.5.2. Temperature Index

The temperature index comprises four temperature parameters: the lower-limit temperature threshold (DV0), lower-limit optimal temperature (DV1), upper-limit optimal temperature (DV2), and upper-limit temperature threshold (DV3). These parameters are typically derived from experimental results and are applied to interpret the setting and distribution data. Duyck and Quilici [28] found that the minimum temperature required for the development of MFF larvae was 10.2 °C by rearing multiple generations of MFF at different temperature gradients indoors. Additionally, the most favorable temperature range for MFF development was found to be 20.6–26.1 °C when rearing MFF indoors [29,30]. Elnagar et al. [31] found that the survival rate of MFF decreased significantly when the indoor rearing temperature was higher than 35 °C. Therefore, DV0, DV1, and DV2 were set to 10 °C, 21 °C, and 26 °C, respectively, while DV3 was set to 35 °C.

#### 2.5.3. Moisture Index

The moisture index comprises the lower limit of upper soil moisture (SM0), lower optimum soil moisture (SM1), upper optimum soil moisture (SM2), and the lower limit of the upper soil moisture (SM3). The MFF primarily inhabits regions around the Mediterranean Sea, known for hot and dry summers and mild, rainy winters. CLIMEX adopts a Mediterranean template, setting SM1 at 0.3, SM2 at 1, and SM3 at 1.5 as references for soil moisture parameters. 

#### 2.5.4. Cold Stress

Cold stress (CS) represents the minimum daily cumulative temperature essential for species survival, determined by the temperature threshold for cold stress (TTCS) and the temperature rate of cold stress (THCS). According to Abd-Elgawad [14], the MFF is unable to continue its development in the non-adult stage at temperatures of 10 °C or below. Therefore, the TTCS was adjusted to 10 °C and the THCS to −0.0007 week^−1^ to account for the coldest regions currently occupied by the MFF, such as France and Hungary.

#### 2.5.5. Heat Stress

The thermal stress temperature threshold (TTHS) and thermal stress temperature rate (THHS) determine heat stress (HS). As HS cannot accumulate within the appropriate temperature range for MFF population development and growth, TTHS must be equal to or greater than DV3 [32]. Thus, TTHS and THCS were set to 35 °C and 0.01 week^−1^, respectively.

#### 2.5.6. Dry Stress

The dry stress threshold (SMDS) and dry stress rate (HDS) delineate dry stress (DS). Due to limited information on the soil moisture requirements of the MFF, parameters were established based on their known distribution. Consequently, the moisture index was derived from that proposed by Kriticos et al. [27], the SMDS was configured at 0.02, and the HDS was established at −0.05 week^−1^.

#### 2.5.7. Wet Stress

Wet stress (WS) is characterized by the wet stress threshold (SMWS) and wet stress rate (HWS), regulating species distribution in conditions of excessive soil moisture. SMWS was defined as 1.5 based on SM3. Given the survival of the MFF in humid regions per distribution records, a low HWS of 0.0015 week^−1^ was set in conjunction with Mediterranean climate template parameters to align the model’s potential distribution with the documented geographical distribution.

#### 2.5.8. Degree Days per Generation

Degree days per generation (PDD) represents the cumulative effective temperature (degree days) above DV0, a critical threshold for completing a generation. According to Abd-Elgawad [14], the effective accumulated temperature of the MFF for a complete generation is 616 degree days; therefore, this we established it at 616 degree days.

### 2.6. Classification of EI Values

The potential distribution of the species was categorized into four groups: unfavorable, slightly favorable, favorable, and highly favorable. To enhance the clarity of the suitability of MFF across different global regions, we adjusted the thresholds for the two suitability categories based on known occurrence records and occurrence density in Europe. The Mediterranean region and southern Africa exhibited relatively high distributions, serving as suitable areas for this pest. We fine-tuned the thresholds until they aligned with the ranges of suitable and highly favorable areas. Critical values for other regions were adjusted according to their occurrence density. The resulting categories were defined as follows: unfavorable (EI = 0), marginally favorable (0 < EI ≤ 10), favorable (10 < EI ≤ 20), and highly favorable (20 < EI ≤ 100).

### 2.7. Mapping the Potential Geographical Distribution and Spatial Variation in PGD

The distribution of suitable zones and four meteorological limiting factors was achieved by applying the inverse distance-weighted interpolation (IDW) method of the spatial analysis module of ArcGIS 10.7 software to interpolate the EI, CS, DS, HS, and WS values of each predicted meteorological data point.

The CLIMEX predictions were imported into ArcGIS and converted to raster files. The ArcGIS reclassification tool was used to classify the suitability into four levels: “unsuitable”, “low”, “favorable” and “highly favorable”. The “Calculate Geometry” tool in the ArcGIS attribute table was used to calculate the area of suitable habitat in the projected coordinate system under different climate scenarios.

Spatial variations in PGD were assessed using ArcGIS 10.7, and the results were categorized as either stable or gaining or shrinking in PGD. 

The arrangement of geographic entities and alterations in the positions of displacement of geographical objects over time can be characterized by their centroids [33]. In this study, the consistency in the shifting of potential habitat areas for the MFF was assessed based on alterations in the duration of potential habitat availability. Initially, the habitat raster map was converted into a vector map using ArcGIS 10.7. Subsequently, the centroid of the PGD was determined using the Statistical Analysis Zonal tool of ArcGIS 10.7.

## 3. Results

### 3.1. PGD of the MFF under Current Climate Conditions

The PGD of the MFF under current climate conditions is illustrated in Figure 2A. The PGD area was approximately 664.8 × 10^5^ km^2^ and was mainly concentrated in Africa, North America, and South America. The highly favorable, favorable, and marginally favorable habitats for the MFF measured 188.6 × 10^5^ km^2^, 178.7 × 10^5^ km^2^, and 297.5 × 10^5^ km^2^, respectively; they accounted for 28.37%, 26.88%, and 44.75% of the total suitable habitat area, respectively (Figure 3). The highly favorable, favorable, and marginally favorable habitats are located in Africa (63.5 × 10^5^, 58.3 × 10^5^, and 44.1 × 10^5^ km^2^), including Morocco, Ethiopia, South Africa, Zimbabwe, Mozambique, Madagascar, and Congo; these areas are followed by southern North America (46.7× 10^5^, 47.3 × 10^5^, and 38.2 × 10^5^ km^2^), including the United States, Mexico, Cuba, Guatemala, and Honduras. Additional habitats were in Asia (20 × 10^5^, 35.5 × 10^5^ and 123.8 × 10^5^ km^2^), South America (46.7 × 10^5^, 47.3 × 10^5^ and 56.6 × 10^5^ km^2^), Europe (11.4× 10^5^, 6 × 10^5^ and 14.5 × 10^5^ km^2^), and Italy, Spain, Portugal, France, Austria, Hungary, and Slovakia, as well as in Oceania (22 × 10^5^, 8.1 × 10^5^ and 20.3 × 10^5^ km^2^). Also included were Australia, New Zealand, and Asia (20 × 10^5^, 35.3 × 10^5^ and 123.8 × 10^5^ km^2^), including China, Vietnam, and Laos (Table 2).

### 3.2. PGD of the MFF under the Future Climate Scenario

The PGD of the MFF projected in future climate scenarios for 2040–2059 is shown in Figure 2B. The total area of the PGD is projected to be approximately 544.1 × 10^5^ km^2^. The potential distribution areas were located in Africa, Asia, North America, South America, Europe, and Oceania (Table 2). The areas of highly favorable, favorable, and marginally favorable habitats for the MFF were 335.2 × 10^5^ km^2^, 101.6 × 10^5^ km^2^, and 107.3 × 10^5^ km^2^ and accounted for 19.72%, 18.67%, and 61.61% of the total suitable habitat area, respectively (Figure 3). Under future climate conditions, the total suitable area for MFF will shrink on all continents except Europe. Compared with other continents, the total area of MFF suitable areas has shrunk the most in South America, being 55.3 × 10^5^ km^2^ (Table 2).

### 3.3. Spatial Variation in PGD under Future Climate Conditions

The assessment of changes in the PGD offers a more intuitive representation of how climate change affects the potential habitats of species. This study characterizes the spatial variations in the PGD of the MFF under future climate conditions, categorizing them as stable, gaining, or shrinking (Figure 4).

The expansion of the PGD under future climatic conditions is mainly projected to occur in North America (northern United States), South America (Argentina), Europe (Germany, Austria, the Czech Republic, Poland, Ukraine, and Russia), and Asia (China, Kazakhstan, and Japan).

Spatial variation classified as shrinking indicates that this habitat will cease to be suitable for *C. capitata* under future climatic conditions. We projected that the habitat of *C. capitata* would be reduced by 120.7 × 10^5^ km^2^ under future climate scenarios, and the shrinking zone is mainly in South America (Brazil), North America (Mexico and the United States), Africa (Botswana, Namibia, Zambia, Guinea, Sierra Leone, Liberia, central Africa, and Somalia), Asia (Turkmenistan, Uzbekistan, Iran, Thailand, and Cambodia), and Oceania (Australia).

The centroids of the PGD of the MFF in each continent in current and future climate scenarios are shown in Figure 5. Under current climatic conditions, the centroids of PGD for the MFF in North America, Europe, Asia, South America, Africa, and Oceania are located in the United States (97.35° W, 30.98° N), Italy (10.95° E, 42.84° N), India (87.37° E, 25.77° N), Brazil (59.62° W, 13.61° S), Congo (23.04° E, 3.07° S), and Australia (136.96° E, 28.46° S), respectively. In 2059, in the future climate scenario, the centroids of the MFF habitats in North America, Europe, Asia, South America, Africa, and Oceania are projected to shift by 418 km, 672 km, 504 km, 686 km, 405 km, and 343 km, respectively, to the United States (98.53° W, 34.82° N), Croatia (18.27° E, 45.35° N), China (90.81° E, 29.27° N), Bolivia (61.13° W, 19.47° S), Congo (25.62° E, 6.08° S), and Australia (139.05° E, 31.02° S). Overall, the centroids of the PGD for the MFF on each continent will shift to higher latitudes in future climate scenarios.

### 3.4. Driving Variables

As shown in Figure 6, heat stress is the main factor affecting the majority of *C. capitata* habitats under current and future climatic conditions. Heat stress affects the distribution in southern and central North America, central Africa, southeastern Asia, central Brazil, and much of Australia, whereas cold stress mainly affects MFF distribution at high latitudes, such as in northern and central North America, Europe, and northern and central Asia. Meanwhile, dry stress mainly affects its distribution in central and southern Africa, southeastern Asia, and northern Oceania, whereas wet stress mainly affects its distribution in northern and central South America, southern and central Africa, and southern Asia.

## 4. Discussion

The MFF is recognized as one of the most formidable fruit fly species [30]. The fruit trade and its innate ability to utilize multiple hosts and adapt to various climates, as well as its high reproductive potential, are the main reasons for its global spread [10,12]. Global warming is an important factor contributing to its spread at high latitudes [34,35]. This study investigated the influence of climate change on the PGD of this invasive species using the CLIMEX model.

Advances in climate modelling techniques depicting a variety of climate scenarios and combining experimental data within modelling have improved the accuracy of predicting changes in the PGD of invasive species [36,37,38]. Over the past few decades, studies have been conducted to simulate the PGD of the MFF using the CLIMEX and MaxEnt models [24,39]. MaxEnt modeling revealed a concentration of suitable habitats for the MFF in southern South America, with North America, particularly the eastern region, being largely unsuitable [39]; this contradicts the results predicted by the CLIMEX model in this study. Differences in the prediction outcomes are mainly due to differences in modeling. The MaxEnt model focuses on determining correlations between the existence of a species and reference points of spatial environmental factors [40]. In contrast, the CLIMEX model uses the biology of the species to set appropriate biological parameters in conjunction with climatic data to predict the range of the fitness zone [41]. The MFF has exhibited a continuing trend of spreading globally [35], which is likely to increase its distribution; therefore, the CLIMEX model is more suitable for predicting its PGD. Our results generally agree with those predicted by Vera et al. [24], who also used CLIMEX. However, they only predicted the PGD of the MFF under current climatic conditions and not its PGD under future climatic conditions. Additionally, we investigated the spatial variation in suitable habitats and the center of potentially suitable habitats for the MFF, which provided better visualization of the alteration of MFF suitability zones under climate change.

Changes in temperature can directly impact the survival and development rates of insects [42], and their geographic distribution can be altered by these changes [43]. Our research suggests that in the future climate scenario, potentially suitable areas and the centroid of potentially suitable habitats for the MFF will shift to higher latitudes. Warming temperatures influence the climate of colder northern regions, making them milder and rendering many currently unsuitable areas suitable for future invasions by the MFF. However, the total PGD area of the MFF is anticipated to decrease. Under future climate conditions, temperatures are expected to rise by 2 °C [44], and a warmer atmosphere can hold more water vapor, leading to a corresponding increase in humidity. Global warming and increased humidity are the main reasons for the reduction of MFF in suitable habitats in the Brazilian region (Figure 6). As the MFF cannot survive temperatures above 35 °C, the extreme heat conditions may result in the death of adults and affect the reproductive capacity of the MFF, decrease the quantity and quality of eggs, and consequently, substantially reduce the PGD within these regions [14]. We also modeled the effects of the CS, DS, HS, and WS indices on the PGD of the MFF under current and future climatic conditions using CLIMEX. Among these, the HS index and WS index are important factors leading to a decrease in the PGD of MFF under future climate conditions (Figure 6). One limitation of the current study is that the CLIMEX model only considers climatic factors [45]. However, aside from climate, additional factors can influence the spread of pests, such as human activities, topographic changes, host plants, and natural enemies [46]. In addition, the metabolic rate of a species may be altered by climate change; therefore, the spatial and temporal variation in parameters is important for describing the reaction of a species to climate change and enhancing the precision of projections [47].

Climate change induces diverse alterations in PGD across different regions. The results of the present study provide a foundation for quarantining the MFF. Regions with habitats that are highly favorable for the MFF should implement strategies to prevent the dissemination of this pest. For example, the southern coastal regions of China, especially the Hainan and Fujian provinces, are not only highly favorable areas for the MFF but also have several important ports of trade, such as the free-trade port of Hainan and the port of Xiamen in Fujian. The spread of this species has been facilitated by increased global trade activities [48], and cities or ports in these coastal countries should strengthen quarantine measures to prevent the introduction of the MFF. Prevention and control measures should be adjusted according to projected spatial variations in this pest under future climatic conditions. Preventive measures should be taken in the expanded areas of PGD, such as Germany and Poland, to prevent its introduction. To combat the further spread of MFF outbreaks globally, the following relevant measures can be taken: insecticide spraying [49], use of protein baiting [50], implementation of biological control methods involving entomopathogens [51], introduction of predators [52] and parasitoids [53], and sterile insect techniques [54].

## 5. Conclusions

The CLIMEX model is based on climatic and biological data and can provide accurate predictions of pest distribution and life cycles. It considers various factors, such as temperature, humidity, and precipitation, making it highly accurate in predicting ecological distributions. In this study, we utilized CLIMEX version 4.0.0 to predict the PGD of the MFF under current and future climate scenarios. These findings suggest that temperature is the primary factor influencing the PGD of the MFF. Under current climatic conditions, the PGD of the MFF is concentrated in eastern and southeastern Asia, southern Europe, southern Africa, central North America, and most of South America. Although the PGD area of the MFF is projected to decrease under future climatic conditions, the centroids of potential distribution on each continent will shift toward higher latitudes, and some European countries, such as Germany, Poland, Ukraine, and the Czech Republic, are at risk of invasion. This study clarified the potential response of the MFF to climate change by 2059 on a global scale. In brief, the impact of the MFF on agricultural development in both its native and invaded regions is likely to be intensified by climate change. The potential range of suitable habitats is projected to extend over a broad latitudinal spectrum. These results may support regional monitoring efforts and decision making by quarantine agencies and trade negotiators.

## Figures and Tables

**Figure 1 biology-13-00177-f001:**
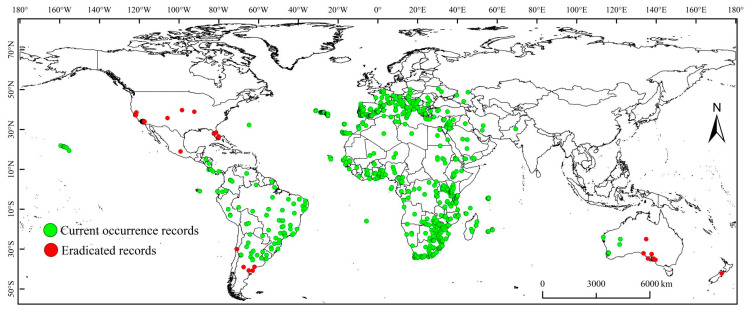
Global distribution of *Ceratitis capitata*. Red dots indicate that *C. capitata* was previously recorded at the site but has now been eradicated, and green dots indicate that *C. capitata* is currently recorded at the site.

**Figure 2 biology-13-00177-f002:**
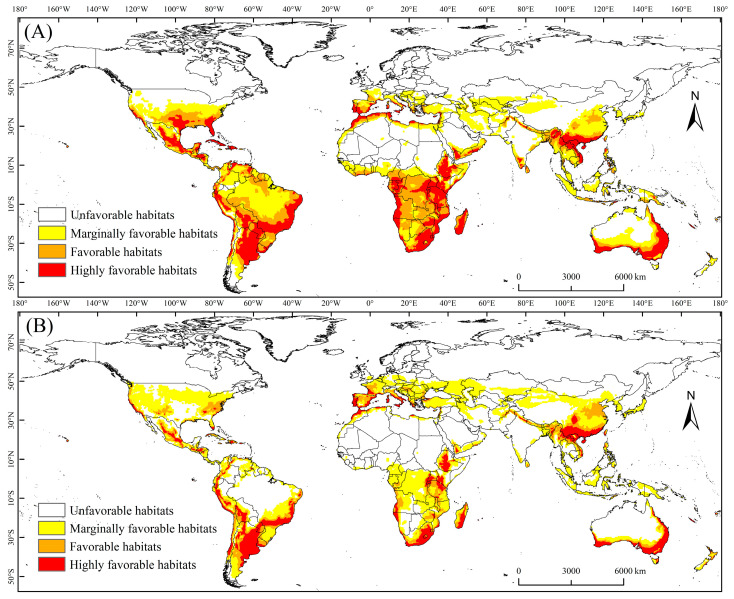
Potential geographical distribution of *Ceratitis capitata* under various climate scenarios. (**A**) Potential geographical distribution under current climate conditions; (**B**) anticipated distribution under future climate conditions (2050s).

**Figure 3 biology-13-00177-f003:**
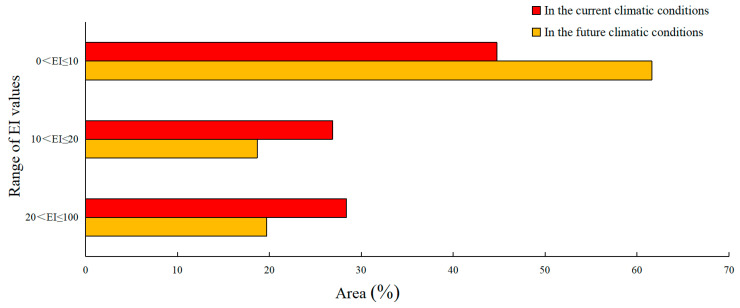
Proportion of different ecoclimatic index (EI) values for *Ceratitis capitata* across various ranges in the habitable zone under both current and future climate conditions.

**Figure 4 biology-13-00177-f004:**
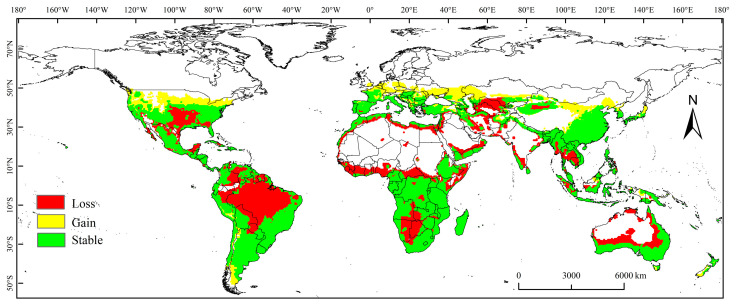
Changes in the potential geographical distribution of *Ceratitis capitata* under future climate conditions.

**Figure 5 biology-13-00177-f005:**
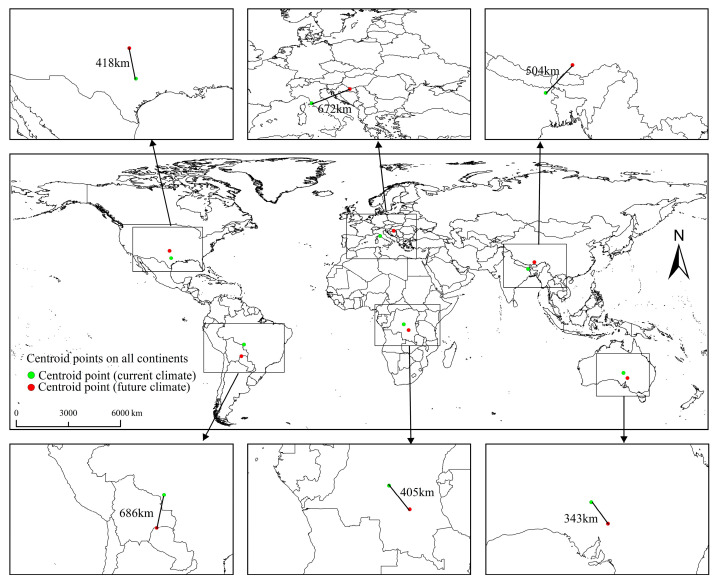
Alterations in the centroids of potential geographical distribution of *Ceratitis capitata* under future climate conditions.

**Figure 6 biology-13-00177-f006:**
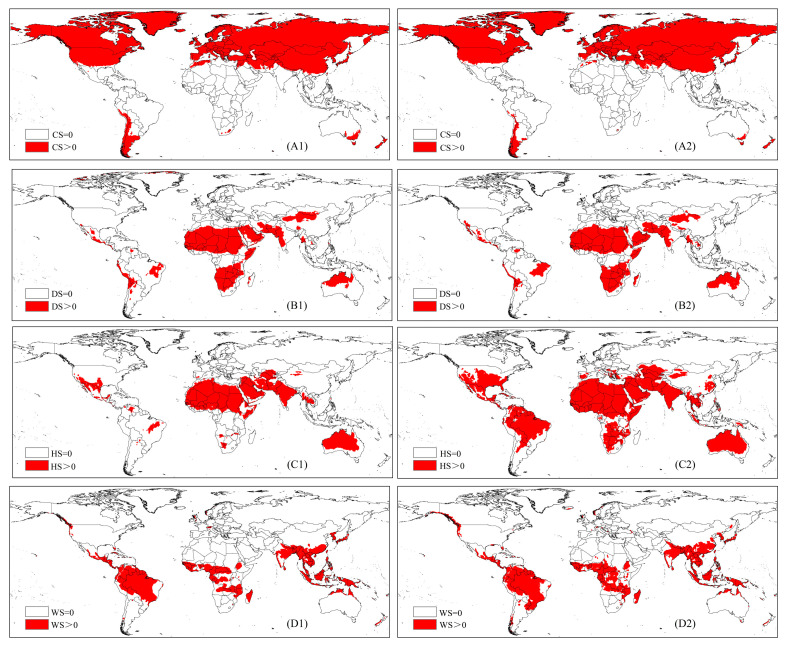
Distribution of four meteorological limiting factors for *Ceratitis capitata*. Red areas in (**A1**,**A2**), (**B1**,**B2**), (**C1**,**C2**), and (**D1**,**C2**) indicate zones unsuitable for *C. capitata* due to cold stress (CS), dry stress (DS), heat stress (HS), and wet stress (WS) indices, respectively. The numbers 1 and 2 correspond to current and future climate scenarios, respectively.

**Table 1 biology-13-00177-t001:** Comparison of model parameter values of CLIMEX for *Ceratitis capitata* by location.

Parameter	Szyniszewska et al. (2020) [26] Parameter Values	Vera et al. (2002) [24] Parameter Values	Final Parameter Values
Lower threshold of soil moisture (SM0)	0.10	0.10	0.10
Lower limit of optimum soil moisture (SM1)	0.30	0.30	0.30
Upper limit of optimum soil moisture (SM2)	1.00	1.00	1.00
Upper threshold of soil moisture (SM3)	1.50	1.50	1.50
Lower threshold temperature (DV0)	10.0 °C	12.0 °C	10.0 °C
Lower optimum temperature (DV1)	21.0 °C	22.0 °C	21.0 °C
Upper optimum temperature (DV2)	26.5 °C	30.0 °C	26.0 °C
Upper threshold temperature (DV3)	35.0 °C	35.0 °C	35.0 °C
Cold stress temperature threshold (TTCS)	10.0 °C	12.0 °C	10.0 °C
Cold stress accumulation rate (THCS)	0/week		0/week
Heat stress temperature threshold (TTHS)	39.0 °C	39.0 °C	35.0 °C
Heat stress accumulation rate (THHS)	0.01/week	0.01/week	0.01/week
Dry stress soil moisture threshold (SMDS)	0.02	0.02	0.02
Dry stress accumulation rate (HDS)	−0.05/week	-	−0.05/week
Wet stress soil moisture threshold (SMWS)	1.6	1.6	1.5
Wet stress accumulation rate (HWS)	0.0015/week	0.0015/week	0.0015/week
Degree days taken to complete one generation (PDD)	662 °C·d	622 °C·d	616 °C·d
Irrigation scenario			1.5 mm day^−1^ in summer

**Table 2 biology-13-00177-t002:** Potential geographical distribution area and percentage of total risk area for *Ceratitis capitata* under current and future climate scenarios.

Continent	Current Climate			Future Climate		
	Highly Favorable Area (10^5^ km^2^)	Favorable Area (10^5^ km^2^)	Marginally Favorable Area (10^5^ km^2^)	Highly Favorable Area (10^5^ km^2^)	Favorable Area (10^5^ km^2^)	Marginally Favorable Area (10^5^ km^2^)
Africa	63.5	58.3	44.1	24.1	23.1	66.7
Asia	20	35.3	123.8	15.2	25.3	123.6
North America	25	23.7	38.2	7.2	17.2	60.4
South America	46.7	47.3	56.6	31.3	22.3	41.7
Europe	11.4	6	14.5	13.6	8.4	33.5
Oceania	22	8.1	20.3	15.9	5.3	9.3

## Data Availability

Data are contained within the article.

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
