# Peer review of "Projecting the Global Potential Geographical Distribution of Ceratitis capitata (Diptera: Tephritidae) under Current and Future Climates"

_biology, 2024, doi:10.3390/biology13030177_

Round 1
Reviewer 1 Report
Comments and Suggestions for Authors
The manuscript “Projecting the Global Potential Geographical Distribution of Ceratitis capitata (Diptera: Tephritidae) Under Current and Future Climates” presents a very interesting exercise to understand how global warming due to climate change may alter the potential global geographic distribution of a serious fruit pest.
The topic has relevance from the basic and applied perspective, particularly to understand how climate change can impact food production through pests shift in their current distributions and which areas or countries increase their risk of introduction and establishment.
The experimental approach by using CLIMEX is adequate. However, there is lack of detail on certain aspects that impedes from assessing the reliability of the interpretation of the results.
My main concerns are associated to:
1. The way the parameters were re-estimated from those already available in the literature. CLIMEX Manual clearly states that parameter fitting is the result of an iterative process and parameters should be taken from experimental data available in the literature as the first step in which validation is necessary. This seems to be missing here.
2. For unfamiliar readers with global warming the mention of a RCP8.5 scenario means little. CLIMEX needs a meteorological data set with specific variables such as temperature, humidity and rainfall. In this sense there is a need to provide more detail on what is changing in the new scenario compared to the current. Is it only temperature or humidity and rainfall are also affected. If so, please provide information on the direction of the change. Global warming is immediately associated with an increase in temperature, in the RCP8.5 scenario are humidity and rainfall increased or decreased
These two aspects are not minor since CLIMEX relies on good parameter fitting and sound climatic data. For the first I have my doubts on the decreased in the DV3 (and consequently increase in the TTHS) and for the second I felt the need of more detail on which is the future scenario.
Along with these major concerns, I provide specific comments in the attached file.
Last but not least, I was surprised in the conclusion when the authors mention cue-lure based attractants when the MFF is not attracted to this lure and when they mention the use or essential oils as a control measure. To the extent of my knowledge this control measure has not been evaluated in the field.
I hope all these comments contribute to a better manuscript, which I would be more than happy to review.

Comments on the Quality of English LanguageI have no particular concerns here but jus minor typo errors such a "C. capitate"
Reviewer 2 Report
Comments and Suggestions for Authors
This is a very good although very common or standard-type (not very novel or original although useful) study nowadays on predicting global distribution of MFF, now and under climate change scenarios. As a result, much of the text reads like so many other papers of this type. For example, the different growth indices are described, here specifically for MFF of course, and justified.
However, there are advancements in methods, so in that regard, the methodologies are not exactly the same as in other papers.
Loss, gain, and stable distribution projections as well as limiting factors analysis add value to the work.
English and grammar need correcting in many places.
capitate should be capitata throughout, but I assume authors know that.
L36 Fruit flies of the family Tephritidae comprise approximately….
L39: capitata
L32: larvae leave holes in fruit that can invite fungi and bacteria
L70: “…predictive instruments..” sounds better
L77: to predict the PGD of tephritid flies.
L123-124: In particular, there have been MFF outbreaks in California in the United States.
I leave other edits such as the above to authors and editor to fix since they do not impact the scientific quality of the work.
Comments on the Quality of English LanguageModerate amounts of editing of English are needed.
Reviewer 3 Report
Comments and Suggestions for Authors
The topic of the paper is very worthwhile and of interest to medfly workers. However, there are large gaps in methodology of the paper that need to be addressed.
line 23 and throughout the paper. How were the climate periods chosen and why is "future climate" specified as 2040-2059, when "current climate" goes from 1981 - 2010? What about 2010 - 2023? And what about 2023-2040?? Without clarity on these periods, the paper makes little sense from the outset.
line 120, 235 and others mention RCP8.5 without any explanation or rationale, though this is a key part of the methodology.
line 100 and Section 2.2. Ecological requirements used in the modeling are evidently based or inferred from the existing MFF distribution. Why didn't the authors search and cite literature on physiological and rearing studies that have determined the tolerances for temperature, humidity, etc? The rearing literature is abundant, needless to say. An example of a study that combines experimental data with field modeling which the authors would do well to cite is:
Rodriguez- Castaneda et al. Journal of Insect Science (2017) 17 (4):88 1-13. doi: 10.1093/jisesa/iex059
line 285. What is "heat suppresion" and why does it occur in future climate. This is just another example of obscure terminology and unexplained methodology that is a serious deficiency of this study.
Reviewer 4 Report
Comments and Suggestions for Authors
Overall a good manuscript. Perhaps I could add a picture of an adult Ceratitis capitata. I recommend the work for publication
Round 2
Reviewer 1 Report
Comments and Suggestions for Authors
The authors have considered most of my previous concerns.
In the attached file you will find three minor aspects to be taken into consideration

Reviewer 3 Report
Comments and Suggestions for Authors
As I have indicated previously, I am satisfied bythe authors' responses to my questionsm, but fail to see these clarifications included in the new manuscript. My interest is not to have just a response letter but to see the responses integrated into the various parts of the manuscript!
